# Cement-Based Repair Materials and the Interface with Concrete Substrates: Characterization, Evaluation and Improvement

**DOI:** 10.3390/polym14071485

**Published:** 2022-04-06

**Authors:** Xuemin Song, Xiongfei Song, Hao Liu, Haoliang Huang, Kasimova Guzal Anvarovna, Nurmirzayev Azizbek Davlatali Ugli, Yi Huang, Jie Hu, Jiangxiong Wei, Qijun Yu

**Affiliations:** 1School of Materials Science and Engineering, South China University of Technology, Guangzhou 510640, China; xuemin1205@163.com (X.S.); 202120120786@mail.scut.edu.cn (X.S.); mshliu@mail.scut.edu.cn (H.L.); jxwei@scut.edu.cn (J.W.); concyuq@scut.edu.cn (Q.Y.); 2Guangdong Low Carbon Technologies Engineering Center for Building Materials, Guangzhou 510640, China; 3Center for Advanced Technologies, Tashkent 100174, Uzbekistan; kasimova_guzal@mail.ru; 4Center for Regulation in Construction, The Republic of Uzbekistan Ministry of Construction, Tashkent 100011, Uzbekistan; aznur90@bk.ru; 5Yellow River Engineering Consulting Co., Ltd., Zhengzhou 450003, China; huangyi121212@163.com

**Keywords:** repair, interfacial microstructure, bonding strength, concrete substrates

## Abstract

Surface damages usually occur in concrete structures. In order to restore the functions and prolong the service life of concrete structures, their surface damages should be repaired in time. This paper reviews the main requirements for repair materials for concrete structures and the most used inorganic repair materials, such as cement-based materials, alkali-activated materials and polymer modified inorganic repair materials. Moreover, techniques to characterize and even improve the interfaces between these repair materials and concrete substrate are summarized. Cement-based material has the advantages of good mechanical properties and consistency with concrete substrate while having the problems of high shrinkage and low flexibility. Polymer modified materials were found as having lower shrinkage and higher flexural strength. Increasing the roughness or humidity of the surface, adding fibers and applying interfacial agents can improve the bond strength between cement-based repair materials and concrete substrates. All of these repair materials and techniques can help to build a good interfacial bonding, and mechanisms of how they improve the interface are discussed in this article. These are of great importance in guaranteeing the effectiveness of the repair of the concrete surface and to guide the research and development of new repair materials.

## 1. Introduction

Cement concrete is currently the most widely used civil engineering construction material worldwide and has been applied in various civil infrastructures, such as roads, bridges, ports and tunnels. Under the external loads and environmental impacts, concrete structures are prone to be damaged, such as cracking and even spalling. In addition, inappropriate selection of raw materials, incorrect mixture design and irregular construction process may also cause damages to concrete, particularly regarding the cover protecting the reinforcement bars. These damages not only reduce the durability of structures, but also lead to structural unserviceability during service [1]. To solve these problems, it is necessary to repair the concrete in time to restore the structures safety and function and to extend their service life.

At present, there is a huge demand for concrete structure repair worldwide. The investments on the repair and reinforcement of old constructions in the USA were estimated at 90 billion dollars in 2016 [2]. In the Middle East Persian Gulf, the expenditure for repair and maintenance in the construction field accounted for 2.6% of the total expenditure in 2011 [3]. In Europe, the budget for maintenance and repair has been close to half of the European construction budget in recent years [4]. In China, the repair and retrofit costs for infrastructure were estimated at 67 billion dollars in 2016 [5]. With the rapid development of civil engineering in the early 21st century, many existing concrete structures worldwide are facing the problem of “aging”. Therefore, demand for building restoration is continually surging.

The effectiveness of surface repair is greatly dependent on the properties of the repair materials and the interface between the repair and concrete substrate [6]. However, the interface between repair materials and concrete substrate is usually a weak zone. There have been various repair materials for concrete structures and plenty of researchers are focusing on the factors that influence the interface and their mechanisms. However, the categories and respective mechanisms are ambiguous, so that high quality concrete structures repair is difficult to achieve. This paper provides a comprehensive review on the existing repair materials and the methods for characterizing or improving the interface between the repair and concrete substrate. Moreover, suggestions for prospective research are proposed.

## 2. Concrete Repair Materials

The effectiveness of repair systems mainly depends on the performances of the repair materials. Usually, repair materials should have good mechanical performances, especially high early mechanical strength, strong bonding with substrates and appropriate workability. The China building materials industry standard (JC/T2381-2016) for repair formulates the fundamental properties (see Table 1) [7]. The European standard (EN1504-3) for repair specifies the relevant requirements for the properties of surface repair materials (see Table 2) [8]. The compressive strength and bond strength are required to be at least 10 MPa and 0.8 MPa, respectively. The American standard (ASTM) for repair also stipulates the relevant requirements for the properties of surface repair materials (see Table 3) [9]. A suitable repair material can effectively prolong the service life of deteriorated concrete structures. At present, the materials for repairing concrete structures have been classified into cement-based materials, polymer materials and polymer-modified mortar [10].

### 2.1. Cement-Based Repair Materials

Cement-based materials are the most widely used materials for repairing concrete structures. These materials not only have a good compatibility with the substrate, but also have advantages of good bonding with the substrate, high economic benefit and easy operations for repair. The most used binders in cement-based repair materials are Portland cement (PC), sulphoaluminate cement (SAC) and magnesium phosphate cement (MPC). In addition, blending materials, such as fly ash, blast furnace slag and silica fume, are usually used to modify the repair materials for better performance.

Qian et al. [11] studied the fundamental properties of cementitious repair mortar and found that the compressive strength of ordinary Portland cement mortar after 1 day curing was 22 MPa and the setting time was over 180 min (see Table 4). In order to shorten the curing time of the repair materials and rehabilitate the performances of concrete structures as soon as possible, the cements used should have a fast hardening property. Compared with ordinary Portland cement, fast-hardening cement has higher early strength because of higher C3S and C3A content [12]. Cifuentes et al. [13] found that the compressive strength of fast-hardening Portland cement concrete at 1 day could reach 42.8 MPa. Compared with ordinary Portland cement, the setting time further decreases from 180 min to 28 min (see Table 4). Therefore, the authors of this study believe that fast-hardening Portland cement is more suitable for emergent repair work [11].

Sulphoaluminate cement is also a fast-hardening binder, of which the main mineral components are C_2_S and 3CaO∙3Al_2_O_3_∙3CaSO_4_. Hydration products with a large amount of crystalline water can not only provide a strong bonding capacity and shorten the setting time for emergent repairs, but also effectively reduce the shrinkage [14]. Yu et al. [15] studied the effect of ettringite on the performance of SAC repair mortar. They found that ettringite seeds can remarkably accelerate the hydration of SAC and increase the compressive strength. When the water-to-cement ratio was 0.28, the SAC mortar strength reached 63 MPa at 1 day, higher than PC mortar, and the final setting time was about 12 min (see Table 5). Qian et al. [11] also reported that SAC mortar exhibited a better early mechanical property than OPC mortar and MPC mortar. However, if the fine needle-like and expansive ettringite was too concentrated and distributed unevenly, the strength of SAC mortar at the later hydration period could decline, which may reduce the quality of repair [16]. Meanwhile, the rapid hardening property of SAC leads to more shrinkage, which may cause the development of cracks. Hajir et al. [17] found that the use of fiber reinforced SAC mortar for the repair of concrete structures can mitigate the influence of shrinkage. They reported that the crack widths in fiber reinforced SAC mortar are 60% lower than the normal SAC mortar.

Magnesium phosphate cement made by mixing the magnesia and phosphate with some inert fillers in certain proportions is a new eco-friendly material because it consumes less resources and energy during production than other traditional cements. Compared to Portland cement mortar, MPC mortar features advantages of higher early strength, shorter setting time, better bonding performance [18,19], lower shrinkage and better environmental tolerance [20,21]. Qian et al. [11] found that the flexural bonding strength between MPC mortar and concrete substrates was about 9 MPa, higher than OPC and SAC mortar, and the setting time of MPC mortar was only 22 min at most, which can significantly shorten the repair period (see Table 6).

Qiao et al. [22] studied MPC with different Mg/P ratio and found that the flexural bonding strength between MPC and concrete substrates reached maximum with Mg/P ratio of 6, and the concrete substrate was 9.1 MPa at 7 days (see Table 6), also much higher than OPC and SAC. Recently, Monica et al. [23] developed an innovative repair material by adding halloysite nanotubes (HNTs) into MPC mortar. It was illustrated that the introduction of HNTs into MPC mortar can improve the consistency and the handling properties of the mortar without affecting the setting time, which is significant, as the fast hardening of MPC is one of the main advantages for concrete repair. Moreover, HNTs can disperse in the MPC mortar and improve the compressive strength. The hydration reaction of MPC is extremely exothermic, making it suitable for applications in cold weather conditions [20,22]. However, when the MPC mortar is exposed to additional water, hydration product MgKPO_4_·6H_2_O dissolves and the porosity of the mortar increases, causing a decrease in the mechanical strength. Therefore, MPC-based repair materials should not be applied in moist or underwater environments [20].

In summary, cement-based materials with the properties of rapid hardening and high early mechanical strength are more suitable for the concrete repair materials. However, the effect of shrinkage of rapid hardening materials should be valued. Among the above cement-based materials, MPC shows better early mechanical properties, shorter setting time and lower shrinkage than PC and SAC.

### 2.2. Alkali-Activated Materials

Alkali-activated material (AAM) is an environmentally friendly repair material because it causes little pollution during production and it presents desirable characteristics, such as high mechanical strength and resistance to chemical agents, better temperature stability, improved abrasion resistance and good adhesion to concrete substrates [24]. Therefore, it has good potential to be used for concrete repair [25]. AAMs are obtained from the alkaline activation of silica (SiO_2_) and alumina (Al_2_O_3_)-rich materials, such as fly ash (FA), blast furnace slag (BFS) and metakaolin (MK) [26]. When AAM is applied to repair concrete substrates, the Ca (OH)_2_ in the substrate will react with the SiO_2_ and Al_2_O_3_ in AAM to form calcium (natrium) silicoaluminate hydrate (C(N)ASH), thereby enhancing the interfacial adhesion [27]. Nunes et al. [28] found that when 20% of MK was partially substituted by BFS, the repair performance was best; the compressive strength was higher than 50 MPa, and the tensile bond strength with concrete substrate exceeded 1.7 MPa (see Table 7). Gomaa et al. [27] found the vertical bond strength between AAM and concrete decreased with the increase in water content. The MK-based AAM featured good workability. The fly ash concrete required the addition of a little expansion agent and plasticizer to reduce the shrinkage of the fresh AAM and enhance its workability [29]. However, the main problem of AAM mortar is the high shrinkage and cracking, which are many times higher than in PC repair mortar. Mariana et al. [24] found that higher aggregate/binder ratio (2.5/1.5), fiber reinforcement and better curing control (prevention of moist evaporation) can significantly reduce shrinkage and cracking.

### 2.3. Polymer-Modified Mortar Repair Materials

Cement-based repair materials usually have problems of high shrinkage and low flexibility. Therefore, modification of cement-based repair materials by using organic materials has attracted lots of attention.

The epoxy-modified cement mortar (EC) has been widely studied for concrete repair [30,31,32] due to its good compatibility with cement-based materials [33,34]. Saccani et al. [35] studied the adhesion and durability of water-based epoxy resin–modified cement mortar used to repair deteriorated concrete structures under high temperatures. The experimental results showed that the EC had good adhesion to the concrete substrate under heating conditions and the mechanical stress caused by differential thermal expansion coefficients. An aqueous epoxy resin was used to modify ordinary Portland cement with four different dosages [36]. It was found that the flexural strength of Portland cement mortar could be improved with epoxy resin reinforcement. It was found that when the epoxy resin was 5% by weight of cement, the mortar has the highest mechanical strength and interfacial bonding strength. (see Table 8). Ariffin et al. [37,38,39] studied the mechanical property of epoxy resin–modified cement mortar with different resin contents and found that that the modified mortars with a resin content of 5% and 10% had compressive strengths of 33 MPa and 36 MPa, respectively, at 28 days. However, when the epoxy resin content exceeded 10%, the mechanical properties of the modified mortar decreased with an increase in the resin content (Figure 1). This can be attribute to the unhardened epoxy resin left in polymer-modified mortars. The excessive epoxy resin inside the mortar probably hinder the hydration and polymerization process. Wang et al. [40] found that the epoxy resin can repair the damaged concrete structure by filling cracks. Due to the good mechanical properties and adhesion of epoxy resin, it can restore the integrity of repaired concrete.

Asphalt mortar (AM) has been applied widely in repairing high-speed railways [41]. Minh et al. [42] studied the factors which influence rheology and hardening properties of AM, such as asphalt–cement ratio, sand–cement ratio and the type and dosage of additive. They developed an AM with high fluidity, rapid setting time and high strength which can improve the stability and regularity of railway roads (see Table 9). Fang et al. [43] found that the interaction between asphalt and cement can be attributed to both physical and chemical actions. Normally, the physical actions of adsorption of asphalt on cement surface and destabilization of asphalt emulsion hold a dominant position. However, when there are more carboxylic acids components in the emulsion, the chemical action, such as interaction with Ca^2+^ and chemical bonding, should be taken in consideration. Liu et al. [44] investigated the bonding mechanism between asphalt-modified SAC repair mortar and concrete substrates. They found that the bond strength between the mortar and substrate decreased with the increase in asphalt content when the asphalt–cement mass ratio exceeded 0.5 (see Table 9), indicating that excessive asphalt had a negative effect on the modified mortar. This can be attributed to the reduction in free water with the increase in asphalt–cement mass ratio, which decreased the amount of Aft, and it is AFt that mainly provides interfacial strength.

Wang et al. [45] studied the flow properties and mechanical properties of cement mortars modified with styrene–butadiene emulsion. It was found that styrene–butadiene emulsion had significant influence on the flexural strength of the cement repair mortar, while it had little influence on the compressive strength. Kharazian et al. [46] found that there is a decreasing trend in the air voids for samples with higher dosage of the copolymer (optimizing at a dosage of 15%). It is further strengthened by the results of mechanical analyses. Bureau et al. [47] found that the strain at rupture and ductility of the mortar increased with the increase in polymer content. Li et al. [48] also utilized styrene–butadiene emulsion-modified cement mortar as a repair material and found that the styrene–butadiene emulsion not only boosted the cement mortar compactness by filling the pores, but also prolonged the mortar setting time because of the retardation effect of polymers on cement hydration. There are two aspects of this retardation effect: (1) The carboxyl groups on the polymer complex calcium ions and (2) polymers are adsorbed on the cement surface and delay the hydration.

In summary, polymer-modified mortar generally has high mechanical strength, excellent durability and high bond strength with the substrates, but its setting time is usually longer than that of the non-modified ones.

## 3. Interface between Cement-Based Repair Materials and Concrete Substrate

A follow-up survey on a large number of existing concrete repair projects showed that most of the repair materials cannot guarantee a long-term effective repair [49]. Among the repair projects considered in the survey, over 90% failed to continue in service for 20 years, and more than half of them were damaged again within 5 years [49]. The failure of repairs in the short term was mainly attributed to the weak interfacial bonding between the repair material and substrate [50].

According to the literature [51], the interface between the repair material and old concrete is similar to the bond between cement paste and aggregates. Therefore, this interface is called the “interfacial transition zone” [51]. The interface is an inherently weak zone, in which the porosity is relatively high and the pore size is large [52]. After the repair material is applied on the substrate surface, a certain incompatibility, inconsistent expansion and shrinkage occur due to the difference in the composition of the two materials (see Figure 2) [11,53,54]. These induce interfacial stress concentration effects and consequently cause interface delamination [53].

Therefore, effective bonding is required to withstand the aforementioned stresses, ensuring the integrity and functionality of repaired structures [54]. The interfacial bonding between repair material and substrate is commonly believed to play an essential role in the long-term effectiveness of the repair [55]. Consequently, the relevant interfacial property and interface formed by the two materials have received increasing attention.

### 3.1. Bonding Strength between Cement-Based Repair Materials and Concrete Substrate

The bond strength between the cement-based repair material and the substrate can be evaluated by means of various testing methods, such as shear bond strength testing (see Figure 3), flexural bond strength testing (see Figure 4), pull-out bond strength testing (see Figure 5) and split-pull bond strength testing (see Figure 6). Flexural bond strength testing is the same as flexural strength testing except for the method of preparing the samples. It can not only determine the flexural bond strength, but also show monolithic failure mode of the repair interface through different fracture positions and sectional conditions [54]. In shear bond strength testing, the samples are fixed and a load perpendicular to the interface is applied. It can determine the shear resistance performance of the repair materials [55]. Pull-out bond strength testing is a direct method to determine the tensile strength. In this test, a square steel block was bonded with the sample surface using epoxy glue, and a vertical pull force was loaded until the interface broke. Split-pull bond strength testing is an indirect tensile test. In this test, the applied force was parallel to the bond line interface of the specimens, and the loading was applied steadily. Then, the split tensile strength can be calculated through the maximum load and bond area. Albidah et al. [56] adopted a core pulling testing method to determine the bond strength between geopolymer repair mortar and old concretes (see Figure 7). In addition to studying the macroscopic mechanical property of interfacial adhesion in concrete surface repair, some scholars have also proposed new methods for testing the bond performance. When the flexural bond strength testing was taken, at the same time, Qian et al. [11] also proposed a new bond performance testing method (see Figure 8) to quantitatively evaluate the impermeability and interfacial bond performance between the repair material and the concrete. The shear–tension strength was determined in this test. A frustum hole was made in the substrate and repair mortar was cast into the hole to approximate half height. When a force is exerted on the upper surface of the frustum in the specimen, shear and tension stress will occur along the interface (see Figure 8). The shear–tension strength can be calculated through the maximum force P. Garbacz [57] adopted a nondestructive impact–echo technique to evaluate the interface quality in a repaired system. For this method, a stress wave that can propagate in the material and have a certain interface reflection was generated by a low-energy impact steel ball and then was detected by piezo electric sensors after propagating through the structure. Accordingly, the amplitude of the obtained spectral peak could be used to reflect the interface bond strength in the repair system (see Figure 9).

Kim et al. [61] carried out pull-out bond strength tests to study the bond performance of cement-based repair materials containing magnesium potassium phosphate for concrete slab repair. Feng et al. [62] conducted split-pull bond strength tests and slant shear bond strength tests to measure the tensile strength and the cohesion and friction components between ultra-high performance concrete (UHPC) repair mortar and normal concrete (NC) substrate. They found that using UHPC to repair the NC substrate led to better bonding performance than NC. This can be demonstrated by the denser and more homogeneous interfacial transition zone between UHPC and concrete substrate.

Through pull-out bond strength testing and double shear bond strength testing, Sadrmomtazi et al. [63] determined the interfacial bond strength of concrete repair layers modified by styrene–butadiene resin-based and acrylic-based polymers, respectively. Zhang et al. [64] conducted pull-out bond strength tests on polymer modified mortar repair and drew a conclusion that the styrene–acrylic copolymer improved the bond strength of the adhesion layer. Julio et al. [65] performed slant shear bond strength tests and pull-out bond strength tests on concrete repair specimens and found that sandblasting on concrete substrate surfaces increased the bond strength between repair and substrate. Tayeh et al. [58] adopted both split tensile and slant shear bond strength tests to study the bond strength between ultra-high performance fiber concrete overlay (UHPFC) and NC substrate. He found that the interfacial bond property of UHPFC and NC substrate performed greatest when the NC substrate had a sand blasted surface because it can promote better adhesion, superior interlocking and, probably, also create a conducive environment pozzolanic reaction of silica fume to take place by increasing surface area. Gomaa et al. [27] adopted oblique shear bond strength tests and pull-out bond strength tests to determine the bond strength between the alkali-activated concrete and ordinary concrete structure. Then, they could also intuitively infer the difference in the interface adhesion according to the failure behavior of the repaired sample.

Through the flexural bond strength tests and the split bond strength tests method, Karima et al. [60] investigated the bond strength and the failure mode of the interface between a special concrete with 100% of limestone filler and concrete substrate. They found that the special concrete had good adhesion to NC. Qin et al. [59] employed the flexural testing method to evaluate the interfacial bond properties of different types of repair materials. Bentz et al. [66] not only performed pull-out and shear bond strength tests, but also carried out interfacial microstructure analysis and neutron photography observation (a technology to display the internal structure of samples by using the attenuation of the neutron beam penetration) to study the influence of the water saturation and surface roughness of the substrates on the bond strength between repair materials and the substrates.

In summary, there are flexural bond strength tests to determine the bend resistance and monolithic failure mode of the repair interface; shear bond strength tests, slant shear tests and push-out tests to determine the shear resistance; and pull-out bond strength tests, split-pull bond strength tests and core pull-put bond strength tests to determine the tensile resistance. In addition, the nondestructive impact–echo technique provides a non-destructive method to evaluate the adhesive property of repair interface.

### 3.2. Characterization on the Interfacial Microstructure between Cement-Based Repair Materials and Concrete Substrate

The interfacial microstructure affects the bond strength between repair materials and substrate. Therefore, it is necessary to characterize their interfacial microstructure.

Zhou et al. [53] characterized the interfacial microstructure between ordinary cement-based repair material and concrete substrate through scanning electron microscopy (SEM). Combining mercury intrusion porosimeter tests and nonevaporating water tests, they found that the moisture transport between cement mortar and substrate affected the interface porosity and the interface bond strength. Through SEM analysis, Gomaa et al. [27] found that a denser microstructure with more calcium aluminate silicate hydration product existed in the interface between alkali-activated concrete and the concrete substrate than in the interface with ordinary concrete. At the same time, Gomaa et al. [27] established the laser triangulation method (which is used to accurately characterize the three-dimensional profiles of the sample, see Figure 10) to quantify the surface roughness of the concrete substrate and collect concrete interface morphology data for determining the relationship between the fractal dimension of the interface and bond strength by using the fractal theory, which was recently used to analyze pore structure of cement-based materials [67,68,69,70].

Qian et al. [11] used SEM and X-ray diffraction technology to study the interface between MPC-based repair material and concrete substrate. They found that the weak acid property brought by phosphate in the repair material induces “the etch” on the substrate surface, so that the material could penetrate into irregular pores and react with the hydration products of Portland cement. The interfacial bonding performance is improved both due to the mechanical interlocking and chemical bonding (see Figure 11 and Figure 12). It can be seen that a higher M/P ratio results in a less condensed microstructure with the formation of less struvite reaction products and more unreacted magnesia grains in the substrate as well as the interfacial zone. Therefore, a loose microstructure appears, as the insufficient hydrates cannot bond all the anhydrous grains. In contrast, the low M/P ratio results in the formation of a continuous struvite phase which can bond the unreacted magnesia grains in MPC mortar and the sand particles and hardened OPC paste in OPC substrate together, acting as an adhesive [59]. Wang et al. [45] adopted SEM to characterize the microstructure of the asphalt mortar–substrate interface. They demonstrated that the high bond strength of such interface was mainly due to the mechanical interlocking between asphalt mortar and substrate. Through mechanical testing and SEM observation, Zhang et al. [71] found that the interfacial bond strength between ultra-high-performance concrete (UHPC, as a repair material) and concrete substrate was high. The interface microstructure and bond strength were greatly improved by calcium-silicate-hydrate (C–S–H) generated by the secondary reaction between the silica fume in UHPC and the Ca(OH)_2_ in the old concrete substrate. Pang et al. [72] used a digital reflective polarizing microscope and SEM to characterize the microstructure of the interface between epoxy-modified mortar and concrete, with nano-silica modified silane as the interfacial coupling agent. It was found that the application of silane-based interfacial coupling agent significantly improved the bond strength of the repair interface because of the reaction between silane and both concrete and epoxy. Moreover, nano-silica modification mitigates the negative effect of dealcoholization of siloxanes on the hydration of the cement.

Sadowski et al. [73] employed a nanoindentation technique to study the micromechanical property of the heterogeneous layer of cement mortar and concrete substrate and to determine the depth of the interfacial zone (see Figure 13 and Figure 14). The regions where indentation modulus (*M*) and hardness (*H*) fluctuate are the interface regions. When the substrate has higher *M* and *H* values, the lowest values of *M* and *H* often appear in the interface region close to the overlay, and the maximum values of *M* and *H* are obtained in the near surface layer of the substrate, which can be attributed to the penetration of the new cement paste of overlay into the substrate.

### 3.3. Technologies for Improving Bond Strength between Cement-Based Repair Materials and Concrete Substrates

The quality of repair is affected not only by the repair material but also by external additives and the concrete substrate to be repaired [74]. Feng et al. [75] proposed that an increase in roughness of the old concrete surface was beneficial to the shear bond strength because of the boost of mechanical interlock. Yazdi et al. [76] applied different surface removal techniques to make rough substrate surfaces and further studied their effects on the bond strength and failure mode of the repair material–substrate interface. The experimental results showed that sandblasting on the substrate surface had the most significant improvement effect on the interface bond strength because of the increase in roughness. Julio et al. [65] drew a similar conclusion. Lukovic et al. [77] found that the water exchange between the substrate and the repair material had an important impact on the interfacial bond property (see Figure 15). When the substrate was pre-saturated, the interfacial bond strength was highest. Courard et al. [78] found that the water content in concrete substrate could significantly affect the cement hydration and microstructure in the repair system due to the moisture exchanged between the repair material and the substrate, and then influenced the interfacial bond strength. The experimental results showed that when the water saturation degree of the concrete substrate’s surface was between 50% and 90%, the interfacial bond performance was best. However, Bentz et al. [66] found that, according to slant shear bond strength tests and pull-out bond strength tests, the bond strengths between the repair and substrate were similar with different humidity of the substrate.

Zanotti et al. [79] studied the effect of steel fiber on the bond strength between cement-based repair material and old concrete. The interface microstructure revealed that the steel fiber in repair mortars can effectively improve the bond strength between the repair material and concrete. As seen in Figure 16 and Figure 17, due to the roughness of the substrate surface, there are many concavities distributed on the interface. Steel fibers settled inside the larger concavities and created a “dowel effect” as they intersected the shear/fracture plane. Feng et al. [62] investigated the adhesion of the interface between concrete and ultra-high-toughness cement-based repair composites. The cement-based repair composite modified by steel fibers had an advantage of significant toughness over the ordinary cement-based repair material. Liu et al. [80] found that a cement mortar, which was modified with a small amount of polypropylene fiber/silica fume and used as a repair material for concrete structure, exhibited improved interfacial bond mechanical properties, which contributed to both the anti-cracking effect from polypropylene fiber and densification reinforcement action to the interfacial transition zones of both fiber–matrix and aggregate–cement paste from silica fume.

To improve the adhesion of the repair material and the concrete substrate, interfacial agents have been applied to the substrate surface. Fahim et al. [81] studied the influence of geopolymer cement slurry as an interfacial agent on the splitting strength between new and old concretes. They concluded that this interfacial agent significantly improved the bond performance between repair material and substrate. Espeche et al. [82] believed that the application of self-compacting micro-concrete made up of a common cementitious mortar (cement, a mix of sand type fine aggregates, a superplasticizer admixture and polymers) as the interfacial agent could reduce the shrinkage of new concrete and generate C–S–H, calcium hydroxide and ettringite in the repaired surface of old concrete. Neves et al. [83] covered a concrete surface with polymer as the interfacial agent and found that the interaction between polymer and hardening cement mortar played a significant role in increasing the interfacial bond strength. Through the bottom-up design for concrete surface, Tatar et al. [84] found that surface functionalization by silane could significantly improve the interfacial fracture energy and durability of cement repair. They suggested that the cohesion between epoxy resin and substrate combined both mechanical interlocking and chemical bonding to improve the interface durability [84]. Xiong et al. [85] found that a silane coupling agent could effectively improve the microstructure of the interfacial transition zone between repair material and old concrete and, consequently, enhance the interfacial adhesion of the repaired layer. Pang et al. [72] applied nano-silica-modified silane as an interfacial coupling agent (SCA) between epoxy-modified mortar and concrete structure. They found that the interfacial agent could improve the bond strength and toughness of the interface between epoxy-modified mortar and concrete structure (see Figure 18).

Based on the literature review above, there are technologies being developed to enhance the interfacial adhesion for concrete repair. Increasing the surface roughness of the concrete substrate and using interfacial agents can improve the interfacial bond strength between cement-based repair and concrete substrate under static loading [81].

## 4. Summary

In this paper, the main requirements for cement-based repair materials for concrete structures were summarized. Mixtures and key properties of some cement-based repair materials and alkali-activated repair materials were introduced.

According to the literature and application experience, the effectiveness of repair of a concrete surface is mainly dependent on the interface between the repair materials and concrete substrates. Therefore, the interfacial properties between cement-based materials and concrete substrate were concentrated on in this review. A comprehensive overview of the methods to evaluate their bond strengths was given. More importantly, characterization of the microstructure of the interface between repair materials and concrete substrates and the techniques to enhance their bond were summarized, as well. Based on the literature review, it can be concluded that the increase in roughness on the substrate surface, the use of polymer-based coupling agents on substrate surfaces and the use of fibers in repair materials can enhance the bonding strength between cement-based repair materials and concrete substrates. In addition, modification of cement-based repair materials with organic materials can also improve their bonding properties with concrete substrates.

However, most of the studies on the bonding between repair materials and concrete substrates were carried out under static loads. It must be emphasized that, in practice, concrete structures usually withstand not only static loads but also repeating loads. Meanwhile, there are usually more pores and defects such as gaps and microcracks in the interfacial zone between repair materials and concrete substrates, which causes stress concentration to easily occur in the interface [62,86,87,88]. Therefore, even though the level of the repeating load is much lower than the material strength, the stress concentration in the interface would promote microcrack formation and propagation and, finally, cause failure. The lack of studies on the fatigue resistance in the repair system for concrete could be a key reason for the absence of durable cement-based repair materials for concrete structures. Insights into the fatigue failure of the surface repair of concrete and methods to mitigate the interfacial stress concentration under repeating loads are key issues in the future investigation for developing durable repair of concrete.

## Figures and Tables

**Figure 1 polymers-14-01485-f001:**
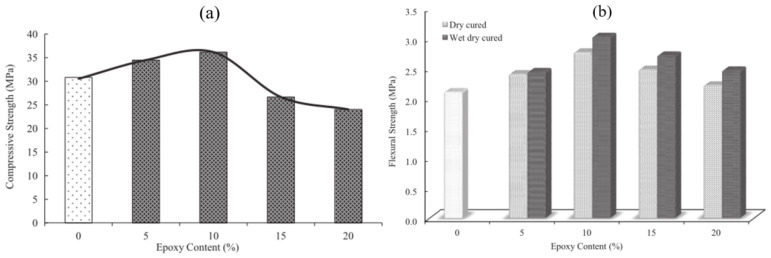
The mechanical strength of epoxy mortar: (**a**) Relationship between compressive strength and epoxy content for wet–dry curing; (**b**) Relationship between flexural strength and epoxy content for wet curing and wet–dry curing [37]. (Adapted and reprinted with permission from ref. [37], © 2015, *Construction and Building Materials*).

**Figure 2 polymers-14-01485-f002:**
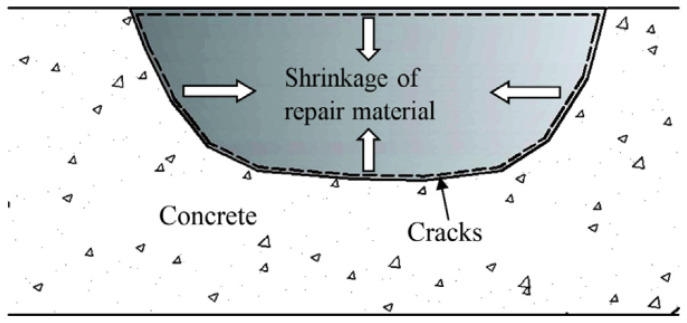
Schematic illustration of shrinkage of repair materials in patch repair [11]. (Adapted and reprinted with permission from ref. [11], © 2014, *Construction and Building Materials*).

**Figure 3 polymers-14-01485-f003:**
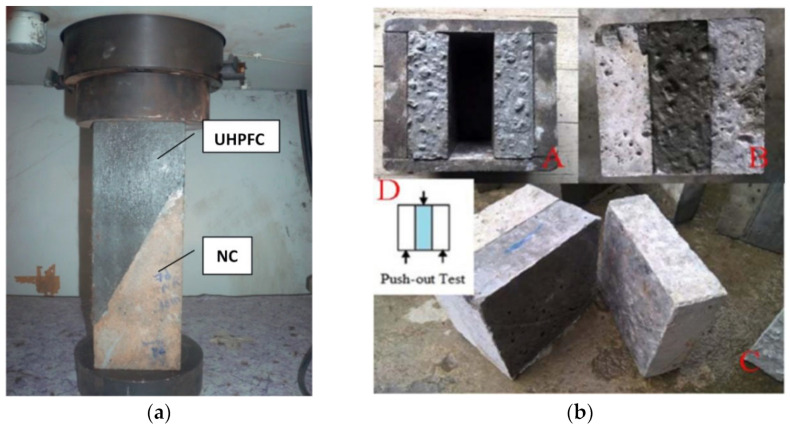
Tests for shear bond strength between repair materials and concrete substrate. (**a**) slant shear test set-up for the composite specimen; (**b**) push-out test for the composite specimen (A, B: preparation of samples; C: the damage mode; D: Schematic diagram of load force) [58]. (Adapted and reprinted with permission from ref. [58], © 2012, *Construction and Building Materials*).

**Figure 4 polymers-14-01485-f004:**
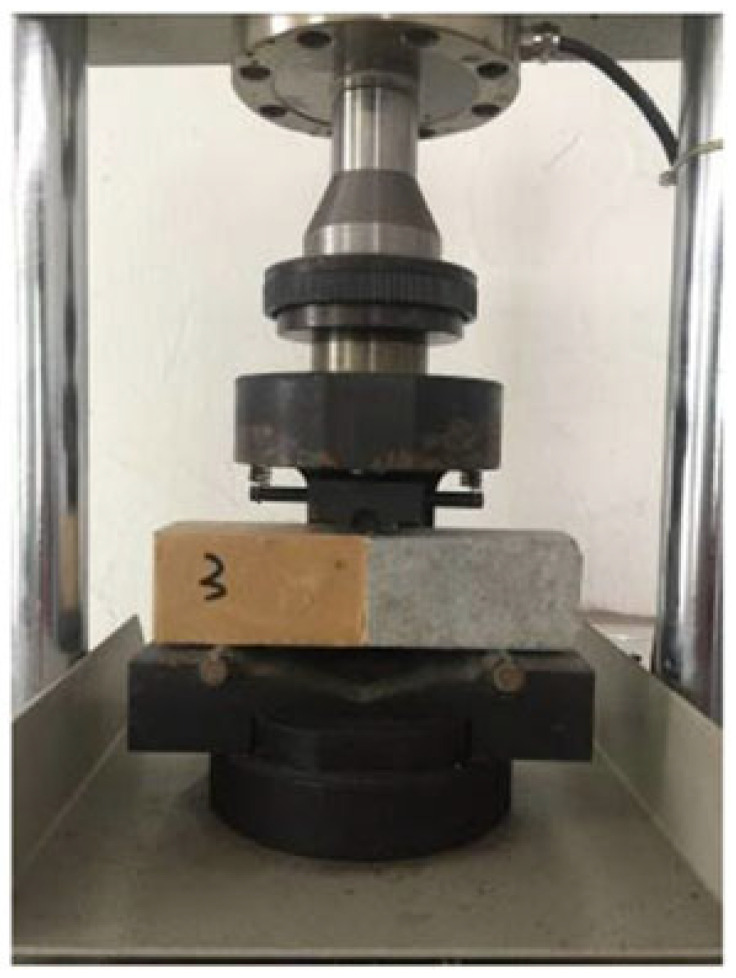
Tests for flexural bond strength between repair materials and concrete substrate [59]. (Adapted and reprinted with permission from ref. [59], © 2018, *Construction and Building Materials*).

**Figure 5 polymers-14-01485-f005:**
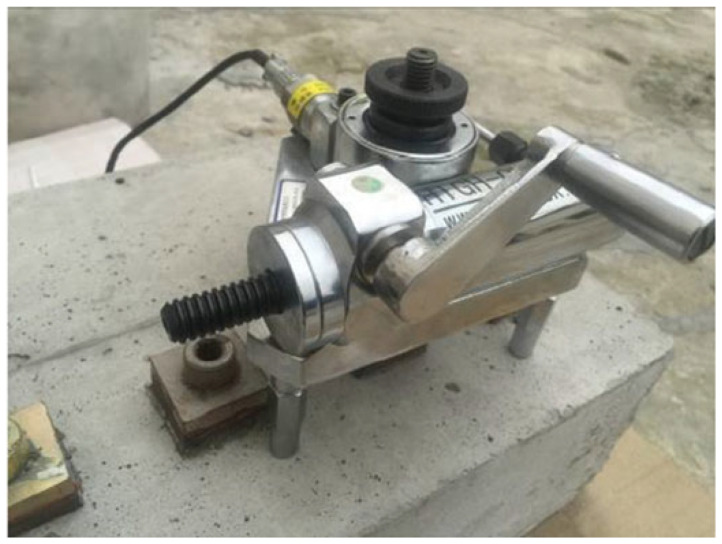
Tests for pull-put bond strength between repair materials and concrete substrate [59]. (Adapted and reprinted with permission from ref. [59], © 2018, *Construction and Building Materials*).

**Figure 6 polymers-14-01485-f006:**
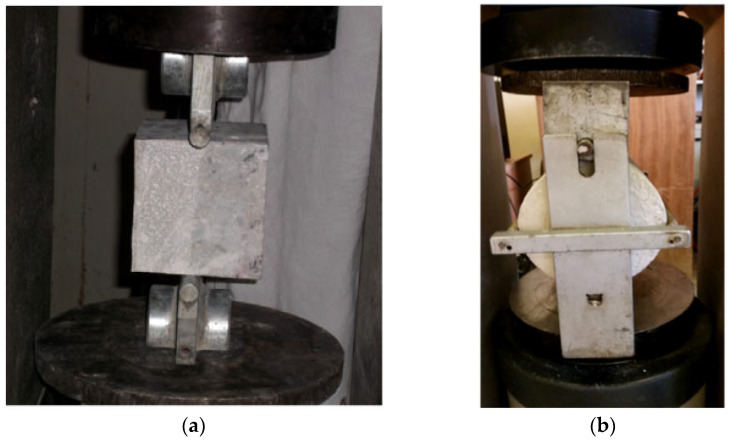
Tests for spilt bond strength between repair materials and concrete substrate. (**a**) cube specimen; (**b**) cylinder specimen [60]. (Adapted and reprinted with permission from ref. [60], © 2017, *Construction and Building Materials*).

**Figure 7 polymers-14-01485-f007:**
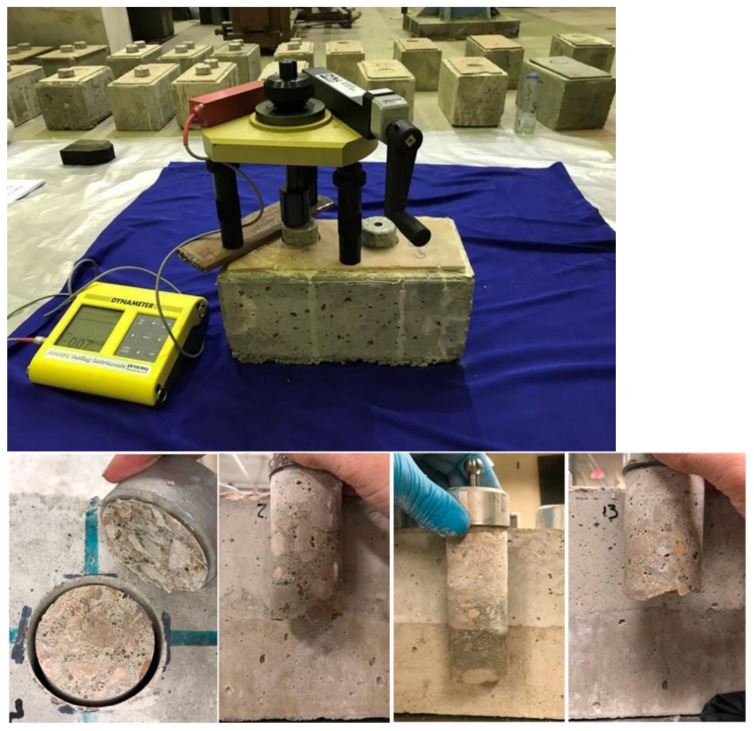
Tests for core pull-put bond strength between repair materials and concrete substrate [56]. (Adapted and reprinted with permission from ref. [56], © 2020, *Journal of Materials Research and Technology*).

**Figure 8 polymers-14-01485-f008:**
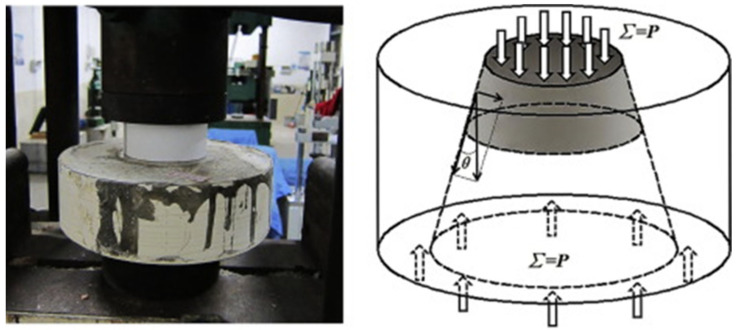
Tests for flexural bond strength between repair materials and concrete substrate [11]. (Adapted and reprinted with permission from ref. [11], © 2014, *Construction and Building Materials*).

**Figure 9 polymers-14-01485-f009:**
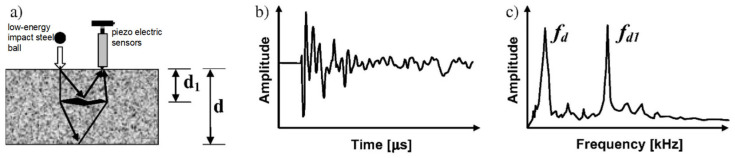
A nondestructive impact–echo technique to evaluate the interface quality in a repaired system: (**a**) Scheme of the impact–echo method; (**b**) example of waveform (time-domain spectrum); (**c**) corresponding frequency when defect in concrete is observed [57]. (Adapted and reprinted with permission from ref. [57], © 2017, *Construction and Building Materials*).

**Figure 10 polymers-14-01485-f010:**
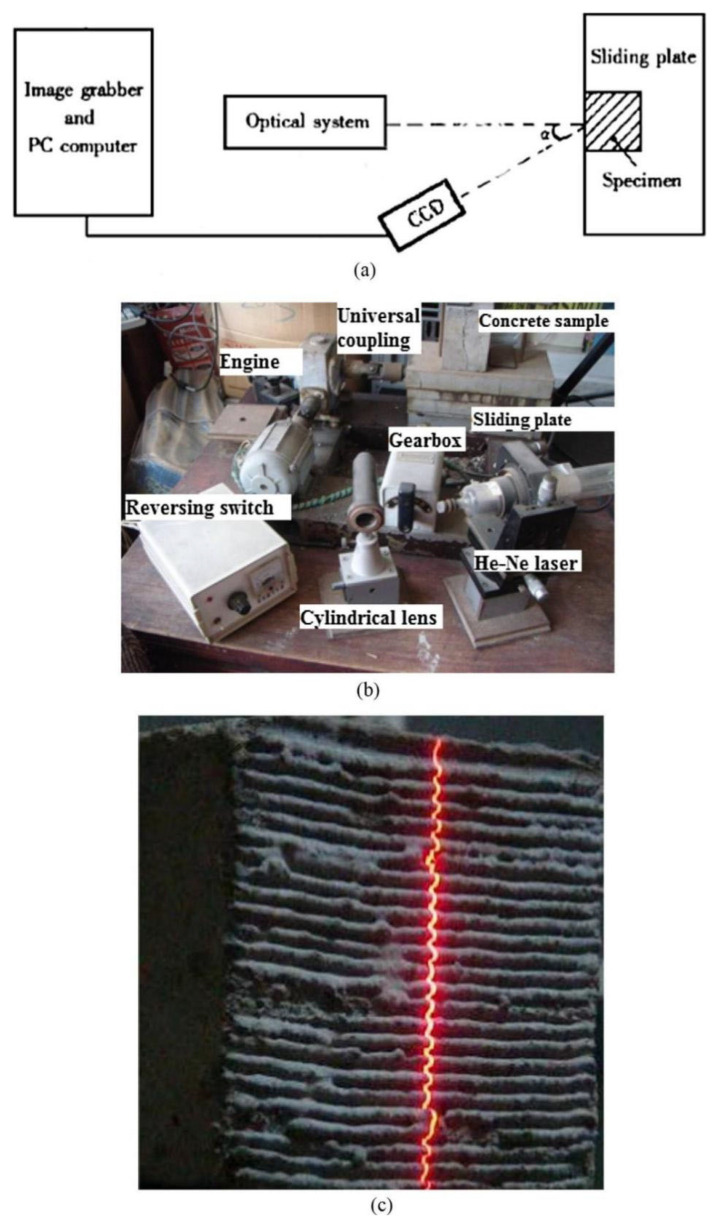
Laser triangulation method to quantified the surface roughness of concrete substrate to be repaired. (**a**) Schematic diagram of laser triangulation ranging test apparatus; (**b**) Installation diagram of laser triangulation ranging test apparatus; (**c**) 3D image of the concrete surface [67]. (Adapted and reprinted with permission from ref. [67], © 2017, *Construction and Building Materials*).

**Figure 11 polymers-14-01485-f011:**
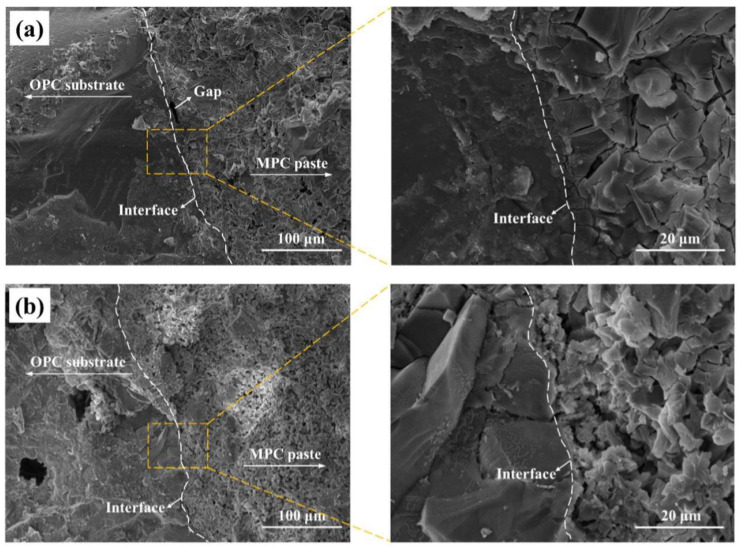
SEM micrograph of the microstructure of the interface (Magnification 500× and 3000×) between old OPC substrate and different MPC matrices with different molar ratios of magnesia and phosphate (7-day-old samples): (**a**) M/P = 10; (**b**) M/P = 14 [59]. (Adapted and reprinted with permission from ref. [59], © 2018, *Construction and Building Materials*).

**Figure 12 polymers-14-01485-f012:**
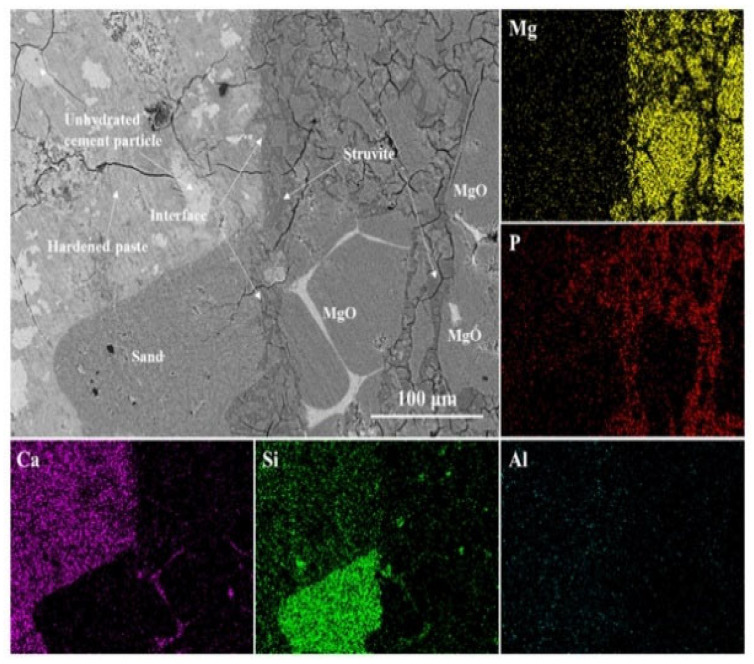
Backscattered electron image and elemental maps of the interface between old OPC substrate and MPC matrix, (M/P = 6) [59]. (Adapted and reprinted with permission from ref. [59], © 2018, *Construction and Building Materials*).

**Figure 13 polymers-14-01485-f013:**
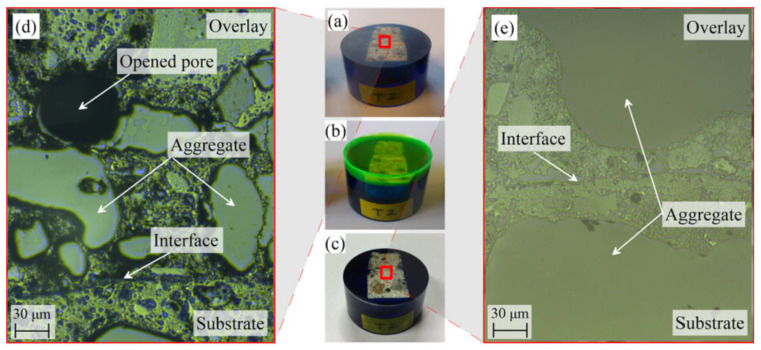
Optical microscopy view of the surface of concrete specimens to be repair. Samples: (**a**) after preliminary grinding (open pores appeared); (**b**) after the second immersion in epoxy resin in a vacuum chamber; (**c**) after final grinding and polishing. The optical microscopy view of the surface: (**d**) after preliminary grinding; (**e**) after final grinding and polishing [73]. (Adapted and reprinted with permission from ref. [73], © 2019, *Construction and Building Materials*).

**Figure 14 polymers-14-01485-f014:**
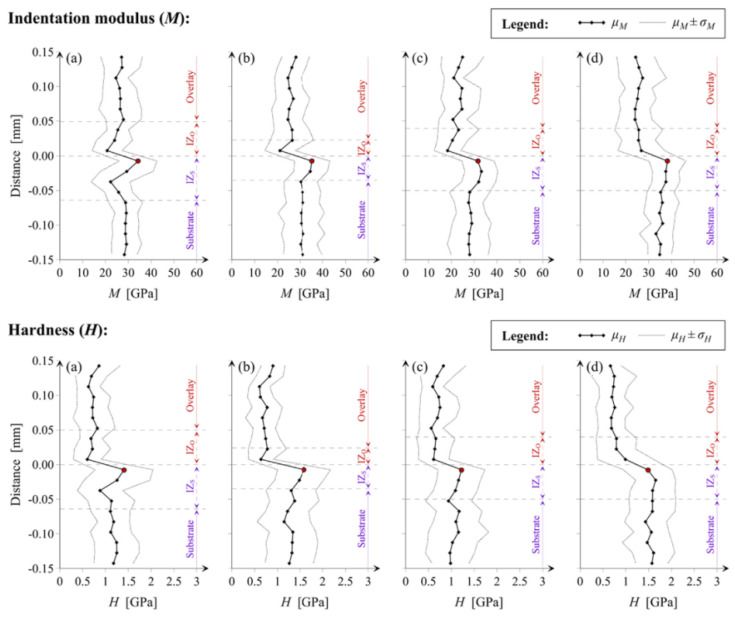
Exemplary maps of indentation modulus (*M*) and hardness (*H*) within the interface between cement mortar and concrete substrate [73]. (Adapted and reprinted with permission from ref. [73], © 2019, *Construction and Building Materials*).

**Figure 15 polymers-14-01485-f015:**
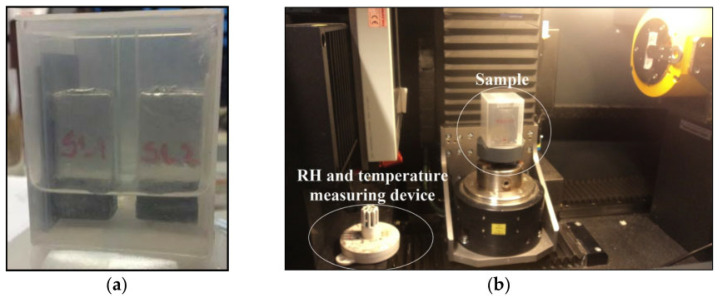
Sample for moisture exchange measurements; (**a**) two specimens are placed in a plastic container pior to X-ray testing; (**b**) the X-ray system with the position of the sample and temperature and RH sensors [77]. (Adapted and reprinted with permission from ref. [77], © 2015, *Materials* (Basel)).

**Figure 16 polymers-14-01485-f016:**
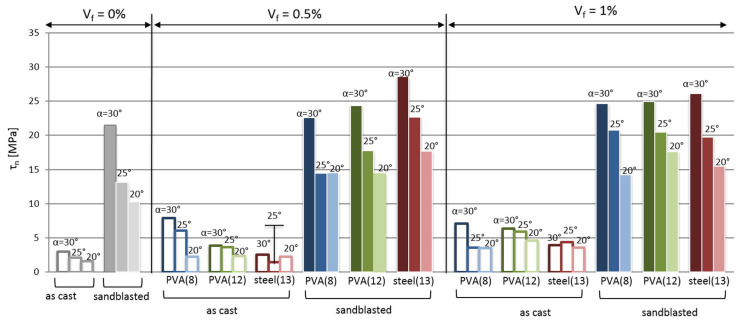
Effect of fiber on the shear bond strength between fiber reinforced concrete and concrete substrate (substrate–repair shear bond strength: τ_n_, plotted as a function of the fiber volume fraction in the repair mortar; V_f_, the bond plane inclination; α, the substrate treatment: sandblasted or “as cast”; the type of fiber reinforcement: PVA or steel (13 mm) [79].) (Adapted and reprinted with permission from ref. [79], © 2018, *Construction and Building Materials*).

**Figure 17 polymers-14-01485-f017:**
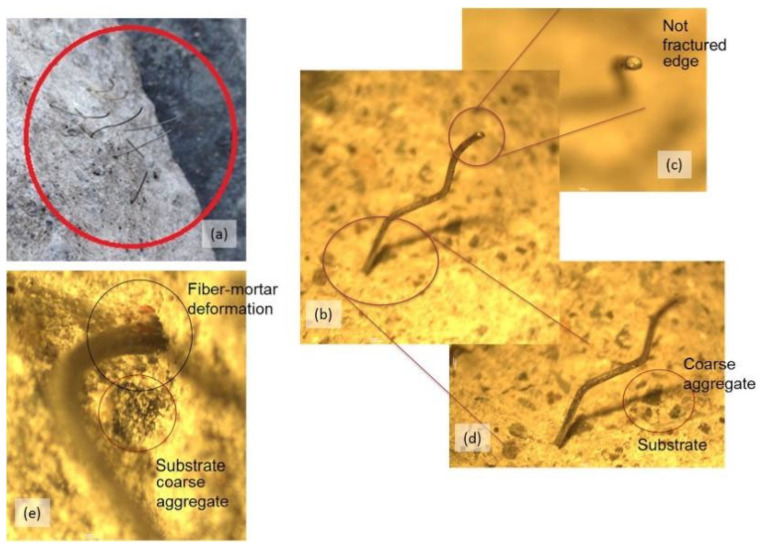
Fracture surface of a sandblasted substrate where steel fibers are bound to repair mortar after failure (**a**) and fiber-concrete details from optical microscope (**b**–**e**). [79]. (Adapted and reprinted with permission from ref. [79], © 2018, *Construction and Building Materials*).

**Figure 18 polymers-14-01485-f018:**
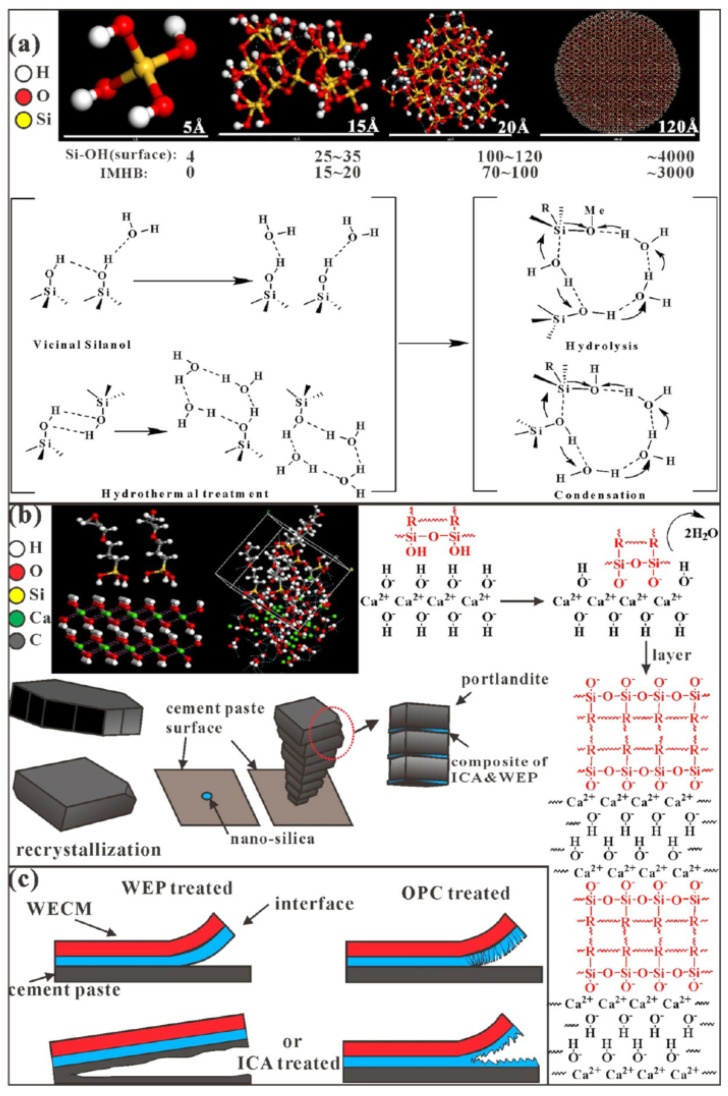
Nano-silica-modified silane as an interfacial coupling agent between epoxy-modified mortar and concrete structure: (**a**) Mechanism of the catalytic effect of hydrothermally treated nano-silica on SCA hydrolysis and condensation, (**b**) fine-grained effect and interfacial coupling effect of SCA on portlandite, (**c**) damage schematic of the interface treated by different means [72]. (Adapted and reprinted with permission from ref. [72], © 2018, *ACS Applied Materials & Interfaces*).

**Table 1 polymers-14-01485-t001:** China Building Materials Industry Standard (JC/T-2016) for the basic performance requirements for repair mortar [7].

Concrete Type	Compressive Strength (MPa)	Flexural Strength (MPa)	Compressive Flexure Ratio	Stretching Bond Strength (MPa)	Rate of Drying Shrinkage (%)
	28 Days	28 Days	28 Days	14 Days	28 Days
NF	≥20.0	≥5.0	≤4.0	≥0.8	≤0.1
NS	≥30.0	≥6.0	≤7.0	≤1.0	

NF: normal flexible repair mortar; NS: normal stiffness repair mortar.

**Table 2 polymers-14-01485-t002:** European standard (EN1504-3) for performance requirements for cement-based structural and non-structural repair products [8].

Item	Requirement			
	Structural		Non-Structural	
	Class R4	Class R3	Class R2	Class R1
Compressive Strength at 7 days	≥45 MPa	≥25 MPa	≥15 MPa	≥10 MPa
Chloride Ion Content	≤0.05%		≤0.05%	
Adhesive Bond at 7days	≥2.0 MPa	≥1.5 MPa	≥0.8 MPa	
Restrained shrinkage Expansion	Max average crack width < 0.05 mm No crack width > 0.1 mm No delamination			No requirement
	≥2.0 MPa	≥1.5 MPa	≥0.8 MPa	
Durability Carbonation Resistance (not required if coated)	d_k_ ≤ Control concrete C (0.45)			Not required
Elastic Modulus	≥20 GPa	≥15 GPa	Not required	

**Table 3 polymers-14-01485-t003:** American standard (ASTM) for performance requirements for repair products [9].

Time	3 h	1 Day	7 Days	28 Days
Compressive strength min, MPa				
R1 concrete or mortar	3.5	14	28	≥28
R2 concrete or mortar	7.0	21	28	≥28
R3 concrete or mortar	21	35	35	≥35
Bond strength min, MPa				
R1, R2 and R3 concrete or mortar	-	7	10	-
Length change, based on length at 3 h max				
R1, R2 and R3 concrete or mortar	allowable increase after 28 days in water			−0.15%
	allowable increase after 28 days in air			+0.15%

B: The strength at 28 days shall be not less than the strength at 7 days.

**Table 4 polymers-14-01485-t004:** Properties of Ordinary Portland cement mortar and Rapid-Hardening cement mortar.

Reference	Mix Proportion	Performance		
	Cement:Sand:Water:Reducer	Setting Time/Min	Compressive Strength at 1 Days/MPa	Flexural Bond Strength at 7 Days/MPa
[11]	1:1.5:0.25:0.01	>180	22	4.8
[13]	1:0.09:-:-	28	42.8	-

**Table 5 polymers-14-01485-t005:** Properties of Sulphoaluminate cement mortar as repair materials.

Reference	Mix Proportion	Performance		
	Cement:Sand:Water:Reducer	Setting Time/Min	Compressive Strength at 1 Days/MPa	Flexural Bond Strength at 7 Days/MPa
[11]	1:1.5:0.25:0.01	15	50	7.8
[15]	1:-:0.3:-	27	40.2	-
	1:-:0.28:-	12	63	-

**Table 6 polymers-14-01485-t006:** Properties of Magnesium Phosphate mortar as repair materials.

Reference	Mix Proportion	Performance		
	Cement:Sand:Water	Setting Time/Min	Compressive Strength at 7 Days/MPa	Flexural Bond Strength at 7 Days/MPa
[22]	1:1.5:0.2 (M/P = 6)	11.25	61.8	9.1
	1:1.5:0.2 (M/P = 8)	8	28	4
	1:1.5:0.2 (M/P = 10)	7.25	37	4.6
[11]	1:1.5:0.12	22	33	4.3

**Table 7 polymers-14-01485-t007:** Properties of Alkali-Active Materials mortar as repair materials.

Reference	Mix Proportion		Performance		
	MK (%):BFS (%)	W/B ^a^	Compressive Strength at 7 Days/MPa	Bond Strength at 7 Days/MPa	
				Flexural Bond Strength	Vertical Bond Stress
[28]	100:0	-	49.3	1.78	-
	80:20	-	50	1.74	-
	60:40	-	56.7	1.3	-
[27]	-	0.36	37	-	26
	-	0.39	33	-	23
	-	0.54	34	-	24.5

^a^ B: Binder; which is either FA or OPC.

**Table 8 polymers-14-01485-t008:** Properties of the epoxy-modified cement mortar as repair materials.

Reference	Mix Proportion	Performance		
	Cement:Sand:Water:Epoxy Resin	Compressive Strength at 7 Days/MPa	Flexural Strength at 7 Days/MPa	Flexural Bond Strength at 28 Days/MPa
[36]	1:2:0.36:0.03	49	9.5	3.8
	1:2:0.36:0.05	50	9.8	4.9
	1:2:0.36:0.07	49.8	9.6	4.7
	1:2:0.36:0.09	49.5	9.4	4.6

**Table 9 polymers-14-01485-t009:** Properties of cement asphalt mortar as repair material.

Reference	Mix Proportion	Performance		
	Cement:Sand:Water:Asphalt Emulsion	Unconfined Compressive Strength at 28 Days/MPa	Setting Time/Min	Flexural Bond Strength at 4 h/MPa
[42]	1:0.5:0.4:0.75	4.1	17	--
	1:0.5:0.4:1	3.4	21	--
	1:0.5:0.4:1.25	1.8	24	-
	1:-:0.4:0.75	2.2	20	-
	1:-:0.4:1	2.1	25	-
	1:-:0.4:1.25	1.7	37.5	-
	Sulphoaluminate cement:sand:water:asphalt emulsion	Unconfined compressive strength at 28 days/MPa	Setting time/min	Flexural bond strength at 4 h/MPa
[44]	1:2.2:0.52:0	-	-	2.65
	1:2.2:0.52:0.1	-	-	2.33
	1:2.2:0.52:0.3	-	-	2.12
	1:2.2:0.52:0.5	-	-	1.87
	1:2.2:0.52:0.7	-	-	0.99

## Data Availability

Not applicable.

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
