# Peer review of "Cement-Based Repair Materials and the Interface with Concrete Substrates: Characterization, Evaluation and Improvement"

_polymers, 2022, doi:10.3390/polym14071485_

Round 1

Reviewer 1 Report

It is a very good study with overall adequate presentation of experimental results. Some additions are needed:

1) Authors should further emphasize on the novelty of their work. It is not clear compared to other published works.

2) Some minor typos, grammar and syntax errors should be carefully revised and corrected accordingly.

3) Reference can be even more updated (more recent relative works).

Author Response

Response 1: 

The novelty and significance have been emphasize in the introduction section.

Response 2: 

The use of English has been carefully revised and improved.

Response 3: 

The reference has been updated by relative articles which are published in 2021 or 2022.

Reviewer 2 Report

Song et al. they present a very interesting and well-written manuscript. I have to congratulate the authors for this manuscript. I would only recommend the authors to improve the way they explain the images. Images should be more self-explanatory. The authors must extensively improve the use of English grammar. This manuscript is good and will be cited.

Author Response

Thanks a lot for your suggestions.

The annotations of images have been revised to be clearer. And the interpretation of the image becomes more detailed. Moreover, the use of English has been revised and improved.

Reviewer 3 Report

The article is about characterization, evaluation and improvement of cement-based repair materials and the interface with concrete substrates. However, some issues must to be addressed:

  1. Abstract: Please start by expressing the aim of this paper, followed by the rest of the information. Typically, the abstract should provide a broad overview of the entire project, summarize the results, and present the implications of the research or what it adds to its field.
  2. The introduction and reference sections must to be improved with newest articles from 2021 and 2022.
  3. The results are merely presented, not properly discussed. Please add explanations for the observed changes. Please give an extended discussion on the obtained results and correlate your findings with previous literature studies and prospective applications.
  4. More analysis and interpretation of the results should be added for a clearer understanding of observed experimental phenomena.
  5. The authors must to provide some details about importance of the research and their applicability.
  6. Please enhance the clarity of the conclusion section in order to highlight the results obtained.
  7. General check-up and correction of the English language is suggested. There are still some minor typos and grammatical errors.

The author needs to address the abovementioned points for the betterment of the manuscript.

Author Response

Thanks a lot for your suggestions.

Response 1:

The abstract has been revised to be more logical.

Response 2:

The introduction and reference have been updated by relative articles which are published in 2021 or 2022.

Response 3:

The explanations of the results became more detailed and more extended discussions have been added.

Response 4:

Further analysis and interpretation have been added to explain the results.

Response 5:

The novelty and significance have been emphasize in the introduction section.

Response 6:

The summary parts have been added to highlight the results obtained.

Response 7:

The use of English has been carefully revised and improved.

Round 2

Reviewer 1 Report

All my comments of the initial submission have been correctly replied and included in the revised manuscript. The quality of this work has been drastically improved after revision and therefore I recommend its publication as it is.

Author Response

Thanks again for your suggestions.  And the English language and style have been further improved.

Reviewer 3 Report

There are improvements on the article and can be published as it is.

Author Response

(The authors gave the same response as above.)
